# Spatial quantification of dynamic inter and intra particle crystallographic heterogeneities within lithium ion electrodes

Donal P. Finegan [1,9*], Antonis Vamvakeros [2,3,4,9*], Chun Tan[5,6], Thomas M.M. Heenan [5,6], Sohrab R. Daemi[5], Natalie Seitzman[1,7], Marco Di Michiel[2], Simon Jacques[3], Andrew M. Beale [3,4,8], Dan J.L. Brett[5,6], Paul R. Shearing[5,6*] & Kandler Smith[1]

The performance of lithium ion electrodes is hindered by unfavorable chemical heterogeneities that pre-exist or develop during operation. Time-resolved spatial descriptions are needed to understand the link between such heterogeneities and a cell's performance. Here, operando high-resolution X-ray diffraction-computed tomography is used to spatially and temporally quantify crystallographic heterogeneities within and between particles throughout both fresh and degraded $Li_xMn_2O_4$ electrodes. This imaging technique facilitates identification of stoichiometric differences between particles and stoichiometric gradients and phase heterogeneities within particles. Through radial quantification of phase fractions, the response of distinct particles to lithiation is found to vary; most particles contain localized regions that transition to rock salt $LiMnO_2$ within the first cycle. Other particles contain monoclinic $Li_2MnO_3$ near the surface and almost pure spinel $Li_xMn_2O_4$ near the core. Following 150 cycles, concentrations of $LiMnO_2$ and $Li_2MnO_3$ significantly increase and widely vary between particles.

[1] National Renewable Energy Laboratory, 15013 Denver W Parkway, Golden, CO 80401, USA. [2] ESRF-The European Synchrotron, 71 Avenue des Martyrs, 38000 Grenoble, France. [3] Finden Limited, Merchant House, 5 East St Helens Street, Abingdon OX14 5EG, UK. [4] Department of Chemistry, 20 Gordon Street, University College London, London WC1H 0AJ, UK. [5] Electrochemical Innovation Laboratory, Department of Chemical Engineering, University College London, London WC1E 7JE, UK. [6] The Faraday Institution, Quad One, Harwell Science and Innovation Campus, Didcot OX11 0RA, UK. [7] Colorado School of Mines, 1500 Illinois St, Golden, CO 80401, USA. [8] Research Complex at Harwell, Harwell Science and Innovation Campus, Rutherford Appleton Laboratories, Harwell, Didcot, Oxon OX11 0FA, UK. [9] These authors contributed equally: Donal P. Finegan, Antonis Vamvakeros. *email: Donal.Finegan@nrel.gov; antony@finden.co.uk; p.shearing@ucl.ac.uk

With falling costs and rising energy density, lithium ion (Li-ion) batteries are becoming the obvious choice for energy storage for an increasing array of applications, with the greatest demand expected to come from electrified transport[1,2]. Improving the performance and safety of Li-ion batteries is imperative. Dynamic chemical and structural heterogeneities across multiple length scales are known to lead to battery degradation and failure[3,4]. For example, strain-induced cracking of electrode particles can stem from lithiation gradients and cause impedance growth[5–8], transition metal dissolution can lead to irreversible capacity loss at both the positive and negative electrodes[9,10], and spatially dependent rates of lithiation can lead to underutilization of capacity[11,12]. Non-destructive in situ X-ray microscopy techniques are valuable tools for quantifying heterogeneities spatially and temporally within cells[8,13–17]. Capturing the dynamics of heterogeneities in large representative volumes, in relevant operating environments, and with resolutions sufficient for sub-particle measurements is highly desirable to achieve insight into inter and intra particle phenomena.

$Li_xMn_2O_4$ (LMO) electrodes are particularly susceptible to capacity fade due to the dissolution of Mn into the electrolyte, its migration and interaction with the positive electrode[9,10,18]. The pursuit of mitigating degradation for LMO is driven by its attractive cost, rate capability, its independence from cobalt, and thermal stability[19–22]. One mechanism of Mn dissolution but perhaps not the only one[23], is that a disproportionation reaction takes place at the interface of the LMO and electrolyte where Mn sites separate into $Mn^{2+}$ and $Mn^{4+}$, where the $Mn^{2+}$ dissolves[23]. Even minor shifts in the oxidation state of Mn in the spinel structure can have consequences for the capacity and capacity retention of the electrode[24,25]. LMO also undergoes numerous phase transitions at different states of charge that are stoichiometry-dependent[21,24–26]. Despite characterization of LMO within Li-ion cells going as far back as the 1980s[19,20] and the extensive neutron and synchrotron X-ray diffraction efforts carried out since[24,25,27–29], a comprehensive spatial and temporal understanding of the chemical and structural heterogeneities and their evolution inside LMO cells, remains elusive[23].

X-ray diffraction-computed tomography (XRD-CT)[30–35] enables non-destructive 3D crystallographic mapping and has been applied to Li-ion batteries for quantifying chemical heterogeneities in the bulk electrode and cell[36–39]. With recent advances in synchrotron brilliance, detector capabilities, and data processing strategies, high-resolution 3D operando chemical imaging is now possible[40]. Representative sample volumes can now be captured with sub-micrometer resolution over short periods of time, facilitating operando, inter and intra electrode particle measurements.

This work demonstrates the application of the state-of-the-art, high-speed and high-resolution XRD-CT capability of the ID15A beamline at The European Synchrotron (ESRF) for characterizing, in 3D, the dynamic crystallographic structure between and within LMO particles during operation. It is shown how the stoichiometry of LMO and its crystallographic response to lithiation varies between particles during operation, and how the stoichiometry of distinct particles changes due to Mn dissolution upon extensive cycling, the extent of which varies widely between particles. This work establishes a major advancement in diagnostic capabilities for complex Li-ion chemistries, which is expected to equip future studies with the tools required for detailing the sub-particle chemical and structural heterogeneities in Li-ion cells for a range of electrode formulations.

## Results

### Electrode characteristics and operando cell performance. The LMO electrode used in this work was 80-μm thick and contained

a wide particle size distribution (Fig. 1) with particle diameters up to ca. 20 μm. Cross-sections of electrode particles in their fresh, uncycled state are presented in Fig. 1a, b where sub-micrometer internal pores are observed. In the fresh state, there are no apparent cracks within the particles. The same electrode was cycled 150 times (see Methods Section for details) and as predicted by Woodford et al.[41], larger particles exhibited a greater tendency to crack, and cracks tended to stem from the edges of internal pores (Fig. 1c, d) as previously predicted[42].

Intra-particle cracks can arise from multiple causes, such as phase transformations, uneven expansion coefficients from the existence of multiple phases, and intercalation-induced stresses[43]. Standard XRD can give some indication that internal stacking faults, secondary phases and change of domain sizes occur over many cycles, by observing broadening of characteristic lattice parameter peaks[29]. To spatially map sub-particle lattice parameters with XRD-CT, a bespoke Li-ion cell with a diameter suitable (1 mm) for high-resolution (1 μm) imaging and low X-ray absorption casing was required. Here, a 1-mm diameter Li vs. LMO cell was seated inside a PEEK housing with terminal pins for compression and to draw current, as shown in Fig. 1e. The design of this cell is discussed in more detail in the Methods section and by Tan et al.[44]. The cell was charged prior to the experiment (see Methods section) and left in a full state of charge (4.2 V). During discharge, operation was intermittently paused for acquisition of diffraction patterns. The discharge profile is shown in Fig. 1f where the blue regions highlight the XRD-CT periods that are labeled from 1 to 5. A slight change in voltage was observed during the XRD-CT periods; this was most likely from charge equilibration or 'relaxation' during open circuit hold[38]. From the integrated XRD point scans taken at the beginning of each pause (Fig. 1g), shifts in the LMO peak positions were observed (i.e. LMO lattice parameter changes during lithiation). $LiMn_2O_4$ adopts a spinel structure, hence lattice constants a, b, and c are equal. The lithiation of $Li_xMn_2O_4$ (where $0 \leq x \leq 1$) occurs in three phases where the lattice parameter ranges from about $a = 8.03$ Å where $x$ is close to 0, to about $a = 8.25$ Å where $x = 1$[24,45,46].

### Operando XRD-CT and electrode heterogeneities. Rietveld refinement of spinel $LiMn_2O_4$ ($Fd3m$) (ICSD: 193444) was carried out on the XRD-CT data sets which were then reconstructed. The fit for the Rietveld refinement was excellent, with lattice parameter error lower than 0.0015 Å (Supplementary Note 1 of Supplementary Information). An initial XRD tomogram with a volume of 202 μm × 202 μm × 40 μm, with 2 μm vertical and horizontal spatial resolutions, was acquired midway through the electrode's depth to identify a region of interest for further high-resolution scans (Fig. 2a). Thereafter, 301 μm × 301 μm × 1 μm XRD-CT slices were acquired close to mid-way through the electrode depth at different stages during discharge of the cell (lithiation of LMO) in which distinct particles could be identified (Fig. 2a). Here, full delithiation where $x \approx 0$ was not reached; the starting phase had a lattice parameter of ca. 8.07 Å, indicating that not all tetrahedral sites were initially vacant of Li which, given the challenges of achieving this state[46,47], was expected.

The XRD-CT slices in Fig. 2a show that there are inter and intra particle lattice parameter heterogeneities, the distribution of which are presented as histograms in Fig. 2b for each of the XRD-CT slices. With knowledge from previously published literature, the shape and evolution of the lattice parameter histograms can provide insight into the phase heterogeneities that existed and evolved upon lithiation. When lithiating, $Li_xMn_2O_4$ is known to undergo two bi-phasic reactions[24,25], the first of which consists of a transition from a phase with lattice parameter $a \approx 8.076$ Å

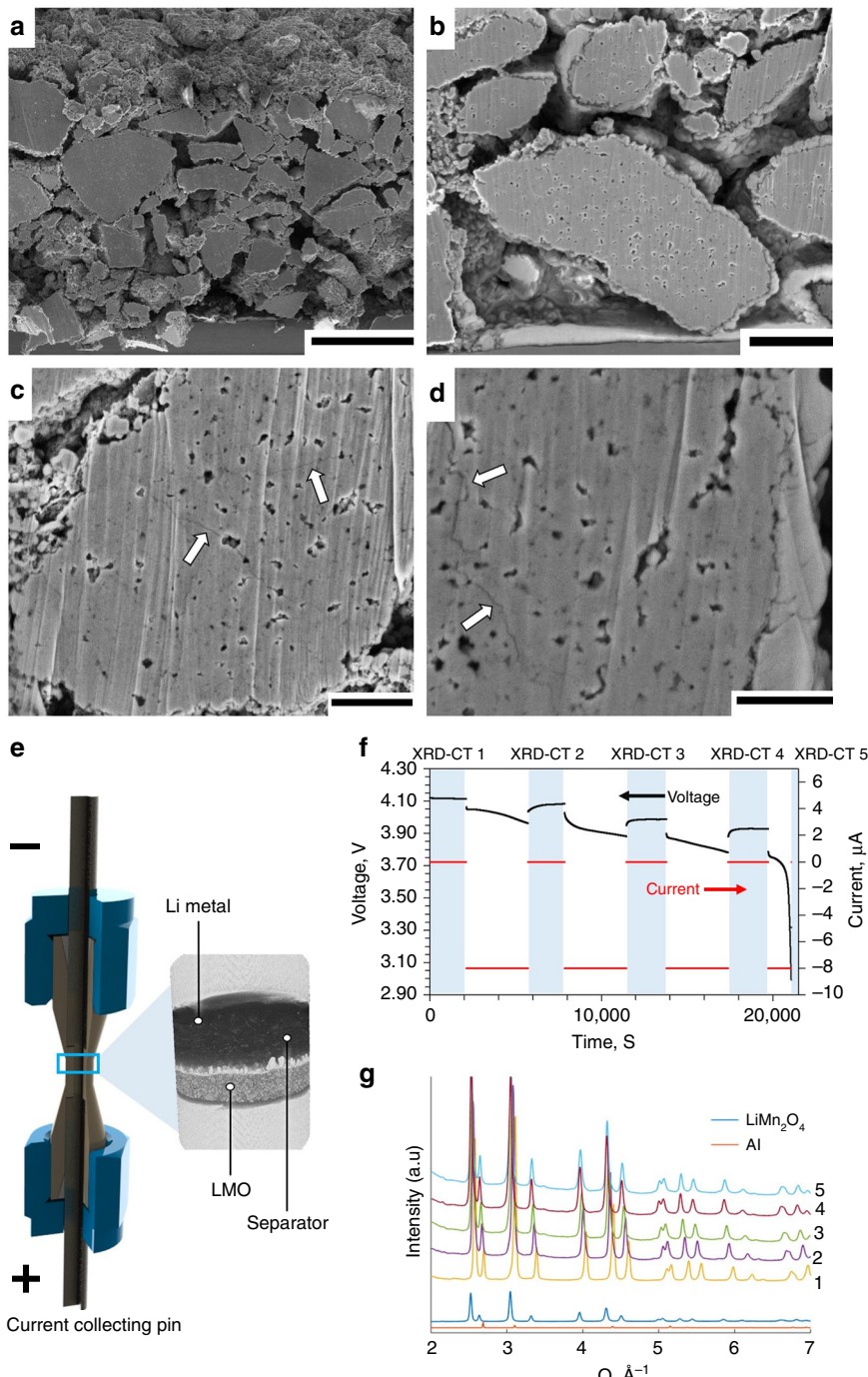

**Fig. 1 Cross-sectioned Li$_x$Mn$_2$O$_4$ electrode and outline of cell design. a, b** Cross-section SEM images of the LMO electrode in its fresh, uncycled state. Scale bars are 25 μm for (**a**) and 5 μm for (**b**). **c, d** Cross-sections of particles that were cycled 150 times, showing cracks propagating from internal pores (white arrows). Scale bars are 2.5 μm for (**c**) and 1.5 μm for (**d**). **e** Illustration showing the design of the operando micro-cell with magnified X-ray CT reconstruction. **f** Discharge voltage and current profile showing periods during which XRD-CT scans were carried out (blue). The XRD-CT periods are labeled 1–5 for future reference. **g** Integrated diffraction profiles from the first point measurement during each of the 5 XRD-CT scans with characteristic profiles from LiMn$_2$O$_4$ and Al for comparison.

directly to a phase with $a \approx 8.145$ Å[24,46]. The second phase change is relatively slight, transitioning from $a \approx 8.183$ Å directly to $a \approx$ 8.205 Å. The lattice parameter range over which the first (larger) phase change occurs is highlighted as pink in Fig. 2b, and is a range that we would not expect to be occupied if the system were pure spinel Li$_x$Mn$_2$O$_4$. However, some volume of the electrode was observed to have occupied this range, thus exhibiting lattice parameter values that did not correlate with the characteristic

behavior of the spinel Li$_x$Mn$_2$O$_4$ stoichiometry. From the sequence of XRD-CT slices in Fig. 2a, the particles that were observed to consistently deviate from the bulk behavior of Li$_x$Mn$_2$O$_4$ were the ones that occupied the non-characteristic range highlighted in pink in Fig. 2b. To understand the discrepancy between the non-characteristic particles and the bulk electrode, single particle analyses were carried out for further insight.

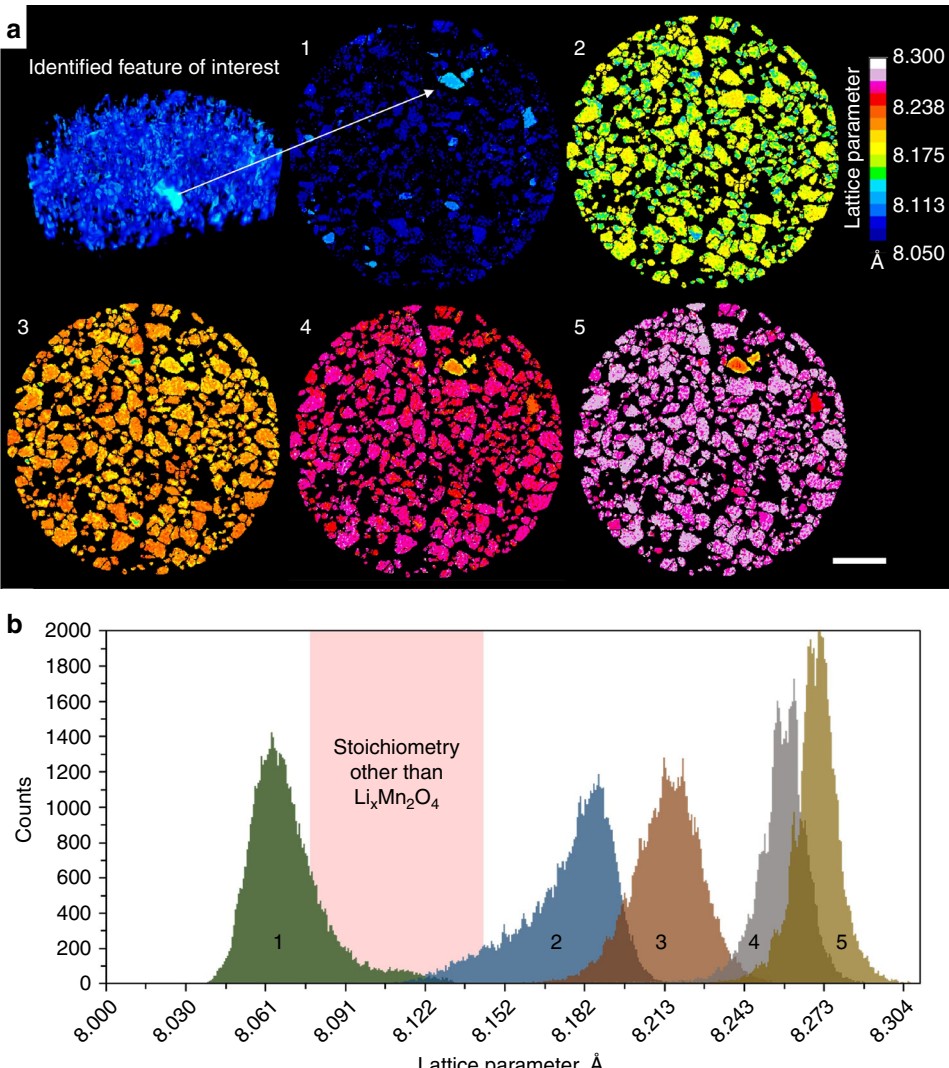

**Fig. 2 Lattice parameters from XRD-CT reconstructions of the Li$_x$Mn$_2$O$_4$ electrode during lithiation. a** (Top left) 2 μm resolution multi-slice XRD computed tomogram used to identify a region of interest. (1–5) Sequential 1 μm resolution XRD-CT slices taken during discharge of the Li vs LMO cell, showing the progression of lithiation of the LMO phase. Scale bar is 50 μm. **b** Histograms composed of the lattice parameter values assigned to each voxel in XRD-CT slices 1–5. The pink region highlights the range of lattice parameter values over which a bi-phasic reaction of Li$_x$Mn$_2$O$_4$ passes without occupying, i.e., a region that is not characteristic of the spinel Li$_x$Mn$_2$O$_4$ stoichiometry.

**Inter and intra particle heterogeneities**. High-resolution XRD-CT facilitated further region of interest investigations of two distinct electrode particles that displayed different crystallographic responses to lithiation. In Fig. 3a, the particle whose behavior diverged most from the bulk (Particle 1), as well as a particle whose behavior was representative of the bulk (Particle 2), were isolated for distinct sub-particle examination.

The histograms of lattice parameter values in Fig. 3b, c show that the lattice parameter of Particle 1, whose behavior most diverged from the bulk, lay squarely in the region where Li$_x$Mn$_2$O$_4$ should not occupy (highlighted in pink). This indicates that the stoichiometry of this particle did not match that of the bulk electrode. Conversely, Particle 2, deemed representative of the bulk (Fig. 3c), exhibited characteristic behavior of the Li$_x$Mn$_2$O$_4$ stoichiometry, where upon lithiation the particle transitioned across the lattice parameter window associated with the larger of the two bi-phasic reactions.

Functional manganese spinels can exist for a wide range of stoichiometries. For example, two groups of stoichiometric derivatives that have been extensively studied are Li-rich[24] stoichiometries conforming to the formula Li$_{1+x}$Mn$_{2-x}$O$_4$ ($0 \leq x \leq 0.33$), or cation-deficient[48] stoichiometries conforming to the formula Li$_{1-x}$Mn$_{2-2x}$O$_4$ ($0 \leq x \leq 0.11$). The oxidation state of the Mn in such stoichiometric derivatives changes from +3.5 for LiMn$_2$O$_4$ and approaches a value of +4, stabilizing the electrode against Mn dissolution for long-life[21]. The electrochemical behavior greatly varies depending on the spinel's stoichiometry, and depending on the value of $x$, a certain number of Li ions may irreversibly replace Mn in the 16d crystallographic sites[21]. Particle 1 in Fig. 3b displays behavior most similar to the Li-rich stoichiometry Li$_{1.10}$Mn$_{1.90}$O$_4$ which, as characterized by Bianchini et al.[24], lithiates through a mono-phasic process beginning with a lattice parameter $a \approx 8.080$ Å and ending with a lattice parameter $a \approx 8.223$ Å. However, in Fig. 3b the histograms from the XRD-CT slices became broader upon lithiation, in particular for XRD-CT slices 4 and 5 where some voxels contained lattice parameter values greater than the expected 8.223 Å. This indicates that sub-particle heterogeneity became more severe with lithiation as quantified in Fig. 4, where Particle 1 and Particle 2 are compared side-by-side.

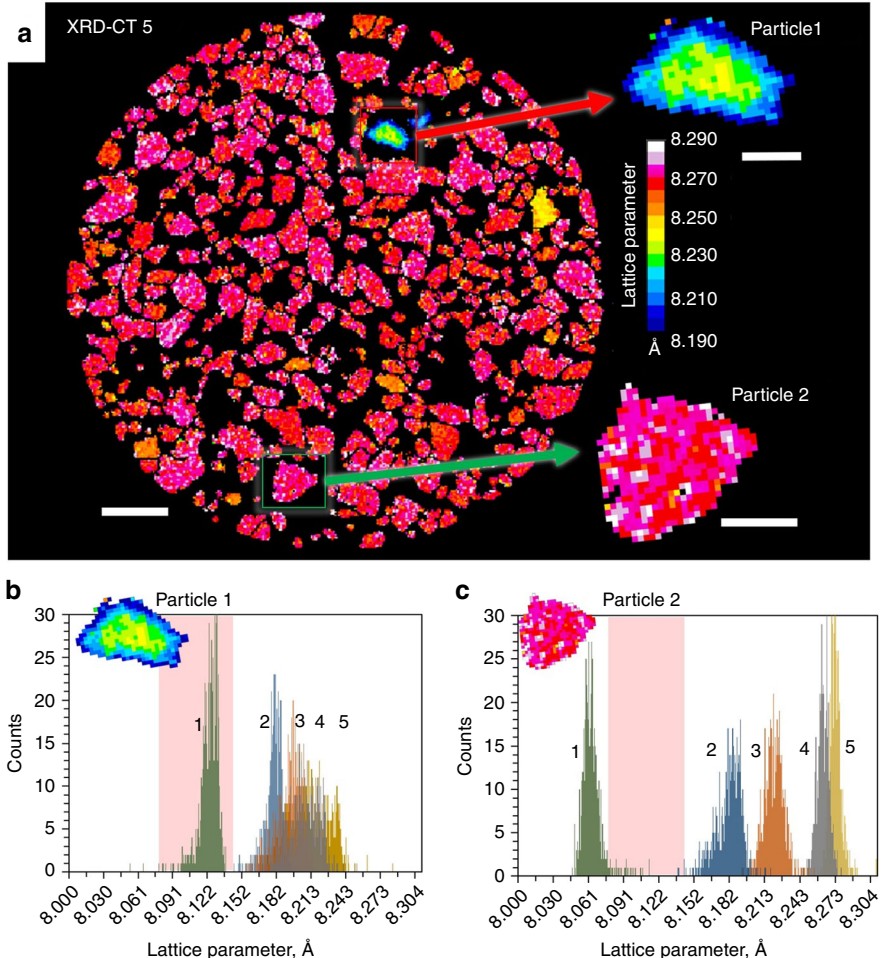

**Fig. 3 Single particle analysis. a** Full view of XRD-CT slice 5 with two magnified particles of interest, one that significantly deviates from the bulk (Particle 1) and one that is representative of the bulk behavior (Particle 2). Scale bar for the bulk electrode is 40 µm and scale bars for enlarged particles are 10 µm. **b–c** Lattice parameter histograms taken from the individual particles in XRD-CT slices 1–5. The pink region highlights the range of lattice parameter values that are not characteristic of pure spinel $Li_xMn_2O_4$.

Particle 1 clearly showed the formation of a lattice parameter gradient within the particle that became more severe as the particle continued to lithiate, whereas Particle 2 showed a uniform lattice parameter profile with radial depth for each state of lithiation. The gradient in Particle 1 consisted of the highest lattice parameter in the center of the particle, for which the most likely explanation is that the particle itself had a slight stoichiometric or phase gradient[49], where with radial distance the characteristic behavior transforms from $Li_{1.10}Mn_{1.90}O_4$ at the surface to $Li_{1.05}Mn_{1.95}O_4$ at the core, suggesting that there was relative Mn deficiency near the surface. This would also explain why some voxels in the histogram in Fig. 3b reached values $a \approx 8.240$ Å, that are in line with stoichiometries between $Li_{1.10}Mn_{1.90}O_4$ and $LiMn_2O_4$[24].

To investigate changes in the crystallographic structure with depth, Particle 1 and Particle 2 were segmented into 4 regions (Fig. 5), to which fitting and Rietveld refinement were applied. The error of the fitting was 0.5 or less (Supplementary Note 1 of Supplementary Information). The mass fractions of distinct crystallographic phases are presented as a function of depth and state of lithiation in Fig. 5. About 3–4% of the material at the surface and subsurface of Particle 1 was shown to be monoclinic $Li_2MnO_3$ (ICSD: 194998) (Fig. 5b), while no presence of $Li_2MnO_3$ was detected beyond the near-surface region. $Li_2MnO_3$ is relatively stable and does not suffer from Mn dissolution,

which makes it ideal as a stable shell phase for 'surface-stabilized' electrodes[22,50]. The gradient of $Li_2MnO_3$ may have formed due to heterogeneous conditions during synthesis[51]. $Li_2MnO_3$ is also a degradation product that forms from dissolution of MnO from $LiMnO_2$, where $LiMnO_2$ forms immediately upon further lithiation of $LiMn_2O_4$;[50,51] but this is unlikely to have been the cause of its presence due to this cell only being cycled once.

The presence of cubic rock salt $LiMnO_2$ (ICSD: 194998) was detected in both particles, but most prominently in Particle 2 following lithiation. The behavior of Particle 2 is akin to the bulk electrode, hence the formation of $LiMnO_2$ in such quantities as observed in Fig. 5d of up to 26%, could have significant consequences for the performance of the electrode. For example, as examined by Yu et al.[52], $LiMnO_2$ has a relatively low-diffusion coefficient and high interfacial reaction barrier thus indicating that the presence of this phase in the bulk electrode shown here may negatively affect its rate performance. In Particle 2, a second LMO phase was detected in XRD-CT 2 at each depth which fitted well with an intermediate phase, $Li_{0.5}Mn_2O_4$ with $P2_13$ space group, recently reported by Bianchini et al.[25]. This intermediate phase disappeared upon further lithiation in Particle 2, and was also detected in Particle 1, but since Particle 1's stoichiometry and phase deviated from the well-documented $Li_xMn_2O_4$, an explanation for its trend with depth and lithiation

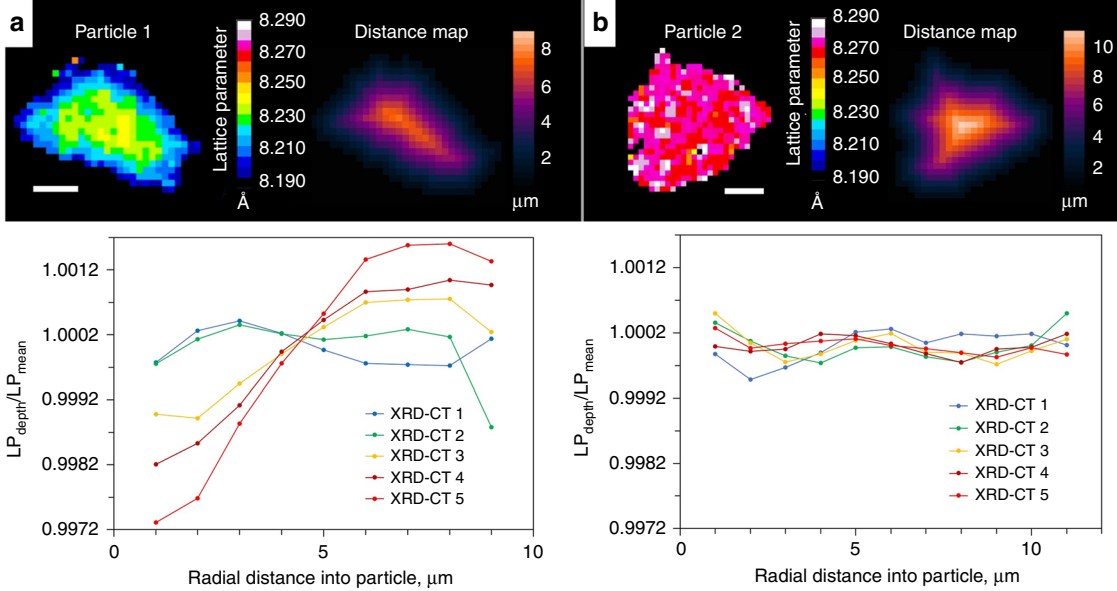

**Fig. 4 Sub-particle radial quantification of lattice parameter. a, b** Top: isolated views of Particle 1 and Particle 2 alongside their corresponding distance maps. Bottom: plots of averaged lattice parameter values for integers of depth in μm's ($LP_{depth}$), divided by the mean lattice parameter of the particle ($LP_{mean}$) for different radial depths into the particles. The radial plots show the formation of **a**, or lack of **b**, chemical gradients that developed upon lithiation. Scale bars are 5 μm.

could not be confirmed. A strain analysis for Particle 2 (Supplementary Note 9 of Supplementary Information) showed that the strain increased with lithiation, which may be due to the phase segregation observed in Fig. 5d as well as the lithiation process itself.

**Spatially quantified phases of degraded electrode.** An LMO vs graphite cell was cycled 150 times, discharged to 3V, disassembled, and packaged in a sealed inert environment for subsequent ex situ imaging. The cell construction and cycling conditions are described in more detail in the Methods section. The capacity fade observed over the 150 cycles is presented in Fig. 6a. An XRD-CT slice was taken from around mid-way through the depth of the electrode. It has previously been shown that degradation can vary with depth into the electrode[53,54], but here we did not focus on depth-dependency due us not expecting significant gradients in lithiation conditions at the low operating rate of C/4. However, we cannot say for certain that the degradation conditions observed for a slice mid-way through the electrode were representative of all depths.

It is shown in Fig. 6b that the bulk of the LMO electrode material had changed into a phase that occupied the range of lattice parameter values that are not characteristic of cubic-spinel $Li_xMn_2O_4$. The spatial distribution of lattice parameters is shown in the XRD-CT slice in Fig. 6b, where distinct particles are seen to have had significantly higher or lower lattice parameter values than the bulk. The low, medium, and high lattice parameter regions were segmented (Fig. 6c) and their respective diffraction profiles analyzed. Three phases were determined, the mass fractions of which are shown in Fig. 6d: spinel $Li_xMn_2O_4$, rock salt $LiMnO_2$, and monoclinic $Li_2MnO_3$. The mass fraction errors were 2 or less for the phases in Segmentation 1, 0.2 for Segmentation 2, and 0.25 for Segmentation 3 (Supplementary Note 1 of Supplementary Information). There were also peaks for other unknown phases, which we were unable to determine.

The formation of $LiMnO_2$ and $Li_2MnO_3$ phases during cycling is well-documented[29,51,55]. Their presence can stem from dissolution of $Mn^{2+}$ from spinel $Li_xMn_2O_4$ and the consequent oxidation of

the residual $Mn^{3+}$ or $Mn^{4+}$ phases. The resulting $LiMnO_2$ and $Li_2MnO_3$ are more thermodynamically stable and co-exist with the $Li_xMn_2O_4$ phase. Migration and deposition of the dissolved Mn on the graphite electrode can cause severe capacity fade through consumption of Li and impedance-rise[9,56]. Here, evidence for Mn dissolution was found by examining the materials at both the positive and negative electrodes. At the graphite negative electrode, X-ray fluorescence confirmed Mn deposits following suspected dissolution and migration of Mn through the electrolyte (Supplementary Note 2 of Supplementary Information)[9]. At the positive electrode, regions with lowest lattice parameter (Segmentation 1 in Fig. 6) were shown to have contained high mass fractions of products that arose from Mn dissolution with 11 % $Li_2MnO_3$ and 20 % $LiMnO_2$. The lattice parameter for the $LiMnO_2$ phase was 4.13 Å, which is similar to that found by Tu and Shu[29]. From Fig. 6, it is seen that the higher the fraction of segregated phases ($Li_2MnO_3$ and $LiMnO_2$), the lower the lattice parameter values observed; this is explained by the fact that as the Mn in $Li_xMn_2O_4$ undergoes disproportionation, all product spinel compounds will have lower lattice parameter values than the original $Li_xMn_2O_4$ phase due to the original spinel having the highest concentration of the relatively large $Mn^{3+}$ ion[51].

With co-existence of phases within single particles, internal strains are expected to have formed. A strain analysis on the three segmented regions in Fig. 6c, d was carried out and is shown in Supplementary Note 9 of Supplementary Information. The strain analysis showed that like Particle 2, strain was highest for regions that had the greatest fraction of segregated phases i.e., highest strain for regions that experienced the highest degree of Mn dissolution. The gained insight into the evolution of sub-particle phase segregation during cycling, and the increase in fraction of segregated phases over many cycles, may help explain the onset of the cracks observed in SEM images in Fig. 1c, d which were taken from the same cycled sample.

**Discussion**
Fast, high-resolution XRD-CT scans taken periodically during lithiation of an LMO electrode facilitated quantification of

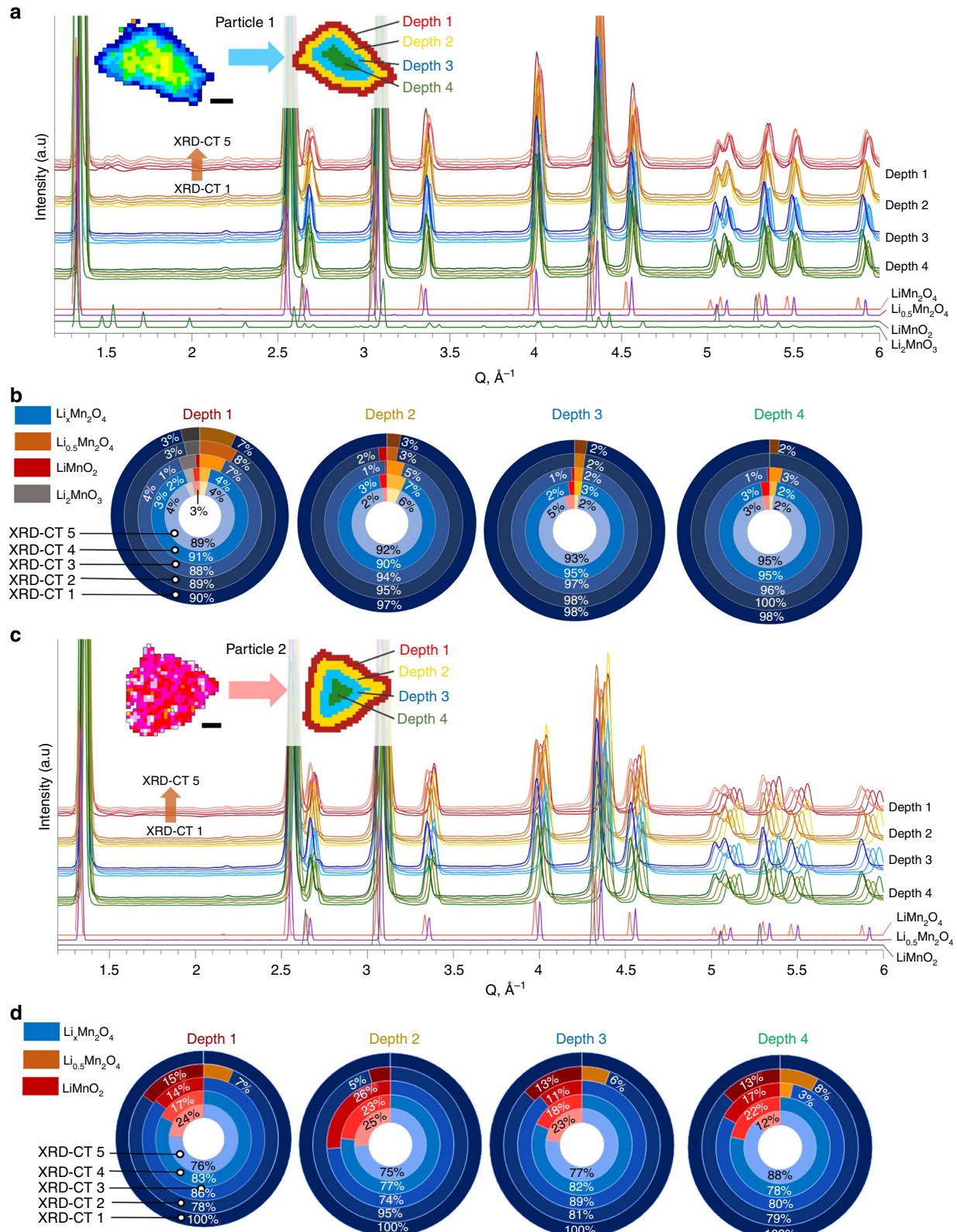

**Fig. 5 Quantification of crystallographic structure with depth. a** Inset showing the segmentation of the XRD-CT slice of Particle 1 into four different depths, alongside the corresponding diffraction profiles for the segmented depths during lithiation from XRD-CT 1 to XRD-CT 5. **b** Doughnut charts showing how the mass fractions of identified phases for each of the 4 depths changed during lithiation. **c, d** The equivalent information for Particle 2. Scale bars are 5 μm.

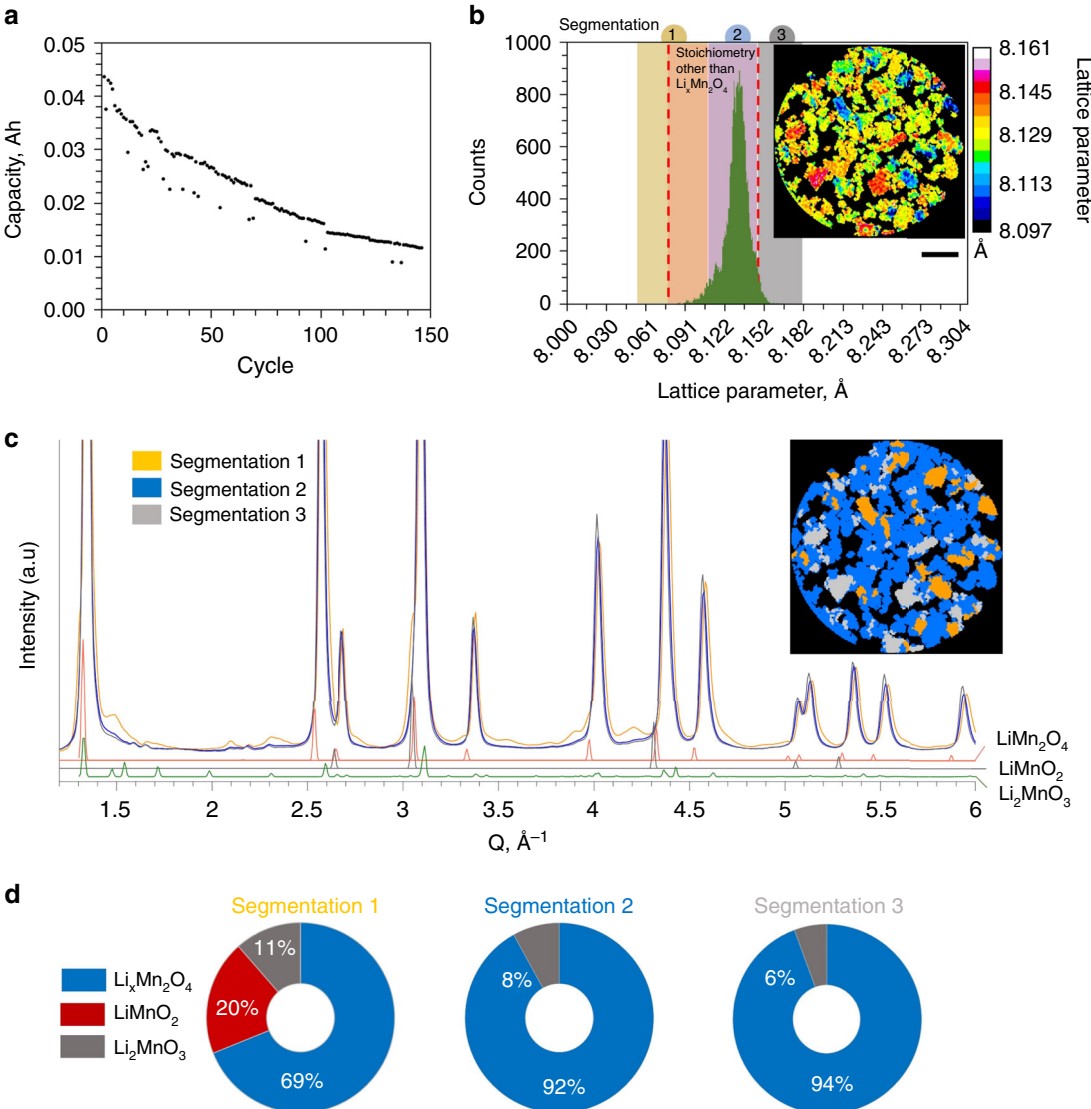

**Fig. 6 Degradation of LMO electrode following cycling. a** Capacity fade observed while cycling the electrode in a $Li_xMn_2O_4$ vs graphite cell for 150 cycles. **b** Lattice parameter histogram for a sample removed from the degraded cell that was discharged to 3V and disassembled. The histogram is overlaid with three ranges of lattice parameter values that were segmented from the inset XRD-CT slice. Scale bar is 50 μm. **c** Diffraction profiles from the 3 segmented regions as shown in the inset segmented XRD-CT slice. **d** Corresponding mass fractions of phases identified in the three segmentations.

stoichiometric heterogeneities between particles and radial variations of stoichiometry and phase mass fractions within single particles. It was shown that the response of distinct particles to lithiation widely varied throughout the electrode. For one particle whose behavior deviated from the expected behavior of $Li_xMn_2O_4$, depth profiling of phase mass fractions revealed a stoichiometric gradient with higher concentrations of $Li_2MnO_3$ near the surface and an abnormally high fraction of an intermediate $Li_{0.5}Mn_2O_4$ phase with a $P2_13$ space-group throughout. In the bulk electrode the intermediate $Li_{0.5}Mn_2O_4$ was identified mid-way through the lithiation process, and rock salt $LiMnO_2$ was shown to form at every depth in most particles during lithiation, even at low states of lithiation during the first cycle of the cell. An LMO sample from the same batch of material was cycled 150 times and imaged ex situ, for which mass fractions of residual phases such as rock salt $LiMnO_2$ and $Li_2MnO_3$, that are known to accrue from dissolution of Mn, were quantified spatially. Their signal was amplified for analysis by segmenting the specific regions where their presence was most prominent. An inverse correlation between the quantity of residual degradation phases and the lattice

parameter of the neighboring $Li_xMn_2O_4$ was identified, such that regions with high amounts of segregated phases displayed lower lattice parameter values for $Li_xMn_2O_4$. The concentration of phases associated with degradation also varied spatially between particles, indicating that not all particles degraded equally.

XRD-CT facilitated amplification of signal from specific phases of interest by segmenting and distinctly quantifying the phase fraction from regions where its presence was highest; using conventional point XRD measurements, such detail would likely be lost in the noise. The ability to distinguish different crystallographic phases as a function of depth into electrode particles during operation is a powerful diagnostic tool for battery research. A plethora of Li-ion electrode degradation mechanisms are associated with radial variations in compositions, for which this technique has been shown to facilitate quantification. This is particularly applicable to next-generation electrodes where the crystal structure and composition is intentionally varied with depth into the particles, such that the core is a high-performance material and the shell a more thermodynamically stable phase that is resistant to transition metal dissolution. Further, the

benefit of combining high-resolution with large sample sizes where heterogeneities between particles are present was also demonstrated, where anomalous particles were identified by comparing to the bulk electrode and distinctly characterized. However, the time required for acquisition of the data presented in this work limited analyses to a single slice and thus prevented statistical volumetric comparisons such as degradation as a function of particle size. The ongoing Extremely Brilliant Source (EBS) upgrade at the ESRF is expected to dramatically decrease acquisition time, and perhaps facilitate operando volumetric comparisons for large samples sizes in future work. This powerful diagnostic tool is expected to become the method of choice for characterizing degradation mechanisms of next-generation Li-ion electrodes as well as other electrochemical energy materials.

## Methods

**Cell materials and cycling conditions.** $LiMn_2O_4$ was the primary cell chemistry of interest for this study. The XRD-CT operando cell consisted of LMO vs. Li, whereas the degraded material for ex situ imaging came from a pouch cell of LMO vs. graphite. The LMO electrode was a commercial single-side $LiMn_2O_4$ coating on a 15 μm thick aluminum current collector (MTI Corporation, Richmond, CA, USA). The graphite electrode consisted of a coating on a 9 μm thick copper current collector (MTI Corporation, Richmond, CA, USA). The $LiMn_2O_4$ electrode was 80 μm thick, had an active material density of 166 g m$^{-2}$ (active material proportion in powder mix was 94.2 wt%) and specific capacity of 110 mAh g$^{-1}$. The graphite electrode was 90 μm thick, had 60 g m$^{-2}$ of active material comprising of 95.7 wt% of the electrode. The specific capacity of the electrode was 330 mAh g$^{-1}$. The pouch cell consisted of electrodes that were 44 × 54 mm and had a capacity of ca. 45 mAh. The electrolyte consisted of 1.2 M $LiPF_6$ in EC:EMC (3:7). The pouch cell underwent formation cycling at constant current 4.5 mA (C/10) between 2.8 V and 4.2 V for two cycles. The pouch cell was charged to 4.2 V at constant current 1.12 mA (C/4), followed by constant voltage until $I < 0.006$ A and discharged to 3V at 1.12 mA (C/4) for 150 cycles. The capacity fade during this time is shown in Fig. 6a. A voltage vs. x in $Li_xMn_2O_4$ profile is presented in Supplementary Note 3 of Supplementary Information.

**Operando micro-cell design and operating conditions.** A cell of Li vs $LiMn_2O_4$ electrode was manufactured inside a Swagelok-union housing design, which will be referred to as 'micro-cell'. The micro-cell consisted of a 1/8" (3.175 mm) PFA Swagelok union fitting shown in Fig. 1e. A bespoke polyether ether ketone (PEEK) body housed the cell (beige in Fig. 1e). Inside the PEEK housing, the cell-chamber was 1.2 mm in diameter. In all, 1 mm disks of $LiMn_2O_4$ and borosilicate glass fiber separator (Whatman GF-D grade, GE) were punched and placed flat on top of a stainless-steel current collecting pin inserted into the cell holder (Fig. 1e). Several drops of electrolyte were added into the cell. Li metal was pressed onto the negative pin, which was then inserted into the assembly displacing excess electrolyte. The Swagelok nuts and ferrules were tightened, making the cell air-tight. The cell was estimated to have a capacity of 0.0143 mAh and underwent a formation charge step at a constant current 4 μA (C/3.6) to 4.2 V before the experiment. The voltage profile during the charge step was as expected and is included in the Supplementary Information. During the operando XRD measurements, the cell was discharged from 4.2 V at 8 μA (ca. C/1.8) until it reached 3 V. A voltage vs. x in $Li_xMn_2O_4$ profile is presented in Supplementary Note 3 of Supplementary Information alongside a typical voltage profile from the pouch cell described above.

**XRD-CT imaging conditions and reconstruction.** The micro-cell was set up to operate while continuously being imaged using XRD. At certain points during charge and discharge, cell operation was stopped and the cell was imaged for XRD-CT. A monochromatic beam of 50 keV (0.2480 Å) was used for all diffraction measurements. The beam was focused to 1 μm using focusing Kirkpatrick-Baez mirrors, and a collimator was placed directly in front of the sample to remove background. A single-photon-counting Pilatus3 X CdTe 2 M detector was used for recording the diffracted signal. The exposure time was 0.01 s for all point measurements.

To reconstruct the XRD-CT slices, three datasets were used: a broad scan that included the PEEK holder, a coarse scan that included the entire width of the electrode, and a region of interest (ROI) scan. The broad and coarse scans were used for correction of the ROI scan by subtracting signal from external entities. The broad scan was taken only once as the PEEK signal would be constant for each reconstruction. The broad scan consisted of 81 rastering points on the horizontal plane separated by 50 μm steps for 80 rotational increments covering 180°.

For the XRD-CT, the coarse scans consisted of 81 rastering points on the horizontal plane separated by 20 μm steps for 81 rotational increments covering a 180° rotation. For XRD-CT of the degraded sample, the coarse scans consisted of 151 rastering points on the horizontal plane separated by 10 μm steps for 120

rotational increments covering a 180° rotation. For the operando XRD-CT, the ROI scans consisted of 351 rastering points on the horizontal plane separated by 1 μm steps for 350 rotational increments covering a 180° rotation. For XRD-CT of the degraded sample, the ROI scan consisted of 251 rastering points on the horizontal plane separated by 1 μm steps for 250 rotational increments covering a 180° rotation. Before the operando XRD-CT scans, an initial XRD-CT volume was captured to identify a region of interest. This consisted of 101 rastering points on the horizontal plane separated by 2 μm steps, 100 rotational increments, for 20 vertical planes separated by 2 μm. Hence a tomogram of 202 μm × 202 μm × 40 μm. This 3D volume was used to identify a region of interest for the higher resolution scans.

A trigger sequence was set up on a Gamry potentiostat where an analog signal was output from the Gamry to the beamline controls to start XRD-CT acquisition when cell operation was paused. During pauses during the cycling sequence (Fig. 1), high-resolution XRD-CT scans were recorded.

Every 2D diffraction image was converted to a 1D powder diffraction pattern after applying a 10% trimmed mean filter to remove outliers using the nDTomo and pyFAI software packages. The data integration was performed with a fast GPU processor[57–60]. The final XRD-CT images (i.e. reconstructed data volume) were reconstructed using the filtered back projection algorithm.

**Preparation of degraded sample.** The degraded pouch cell was discharged to 3 V and dismantled inside an argon-filled glove box. The electrodes were washed with dimethyl-carbonate and allowed to dry. The 1 mm diameter disks were punched out and glued to the top of a steel pin for imaging. The samples were sealed in an argon-filled bag until they were ready to be imaged.

**Rietveld refinement and data processing.** More information on the reconstruction process is provided in Supplementary Note 4 of Supplementary Information. First, a mask was applied to a coarse XRD-CT dataset that contained the casing material of the cell. This was to extract a diffraction pattern containing only signals generated by the PEEK component of the casing. For the high-resolution region-of-interest XRD-CT data, the scattering/diffraction signals from components other than the electrode were absent, and the model included only the $LiMn_2O_4$ phase unless stated otherwise. Apart from the background and the scale factors for all phases, the other parameters refined during the Rietveld analysis of the diffraction data were the lattice parameters for $LiMn_2O_4$. The analysis of the diffraction data was performed with the Topas v5 software[61]. The steps taken to segment the particle phase, identify phases including the secondary $Li_{0.5}Mn_2O_4$ phase, isolate the two distinct particles for comparative analysis, and identify the cubic rock-salt $LiMnO_2$ phase, are discussed in detail in Supplementary Notes 4 to 8 of Supplementary Information.

## Data availability

All data presented in this manuscript are available from the corresponding authors upon request.

## Code availability

The code used to reconstruct the data in this manuscript is available from the corresponding authors upon request.

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

## Acknowledgements

D.P.F. would like to acknowledge help from Jeff Gelb (Sigray Inc.) in fluorescence measurements of degraded electrodes. D.P.F. would also like to thank Mowafak Al-Jassim and Harvey Guthrey for allowing us to use the FIB-SEM system. These experiments were performed at beamline ID15A at the ESRF (Grenoble, France). We are grateful to the ESRF for allowing us to use their facilities. This work was authored by Alliance for Sustainable Energy, LLC, the manager and operator of the National Renewable Energy Laboratory for the U.S. Department of Energy (DOE) under Contract No. DE-AC36-08GO28308. Funding provided by the U.S. Department of Energy Vehicle Technology Office. The views expressed in the article do not necessarily represent the views of the DOE or the U.S. Government. The U.S. Government and the publisher, by

accepting the article for publication, acknowledges that the U.S. Government retains a nonexclusive, paid-up, irrevocable, worldwide license to publish or reproduce the published form of this work, or allow others to do so, for U.S. Government purposes. Antonis Vamvakeros is supported (in part) through funding received from the European Union Horizon 2020 research and innovation program under Grant Agreement No. 679933 (MEMERE project). P.R.S. acknowledges funding from the Royal Academy of Engineering (CiET1718\59) and The Faraday Institution under ISCF Faraday Challenge Fast Start project on "Degradation of Battery Materials" made available through grant EP/S003053/1.

## Author contributions

D.P.F., A.V., and P.R.S. conceived, prepared, and conducted the experiments. A.V., C.T., T.M.M.H., S.R.D. and M.D.M. helped with carrying out the beamline experiments. M.D.M. prepared the beamline and X-ray conditions for the experiment. N.S. carried out FIB-SEM of the electrode samples. A.V., S.J. and A.M.B. made the software for reconstructing the XRDCT data. D.P.F., A.V., S.J., A.M.B., D.J.L.B., P.R.S. and K.S. helped interpret the data and provided guidance on writing the manuscript. D.P.F. and A.V. processed the data and wrote the manuscript.

## Competing interests

The authors declare no competing interests.
