## [Peer Review File · Nature Communications]

Reviewers' comments:

Reviewer #1 (Remarks to the Author):

The article is based on solid experimental results consisting of high resolution XRD-CT data. The data provide the framework to understand the phase change and degradation mechanisms in LMO cathode particles and can be extended to other material systems. In this respect, the manuscript is valuable. However, I have some reservations regarding its publication in Nat-Comm as mentioned below.

1. The logical argumentation which provides the extension of experimental observations to the insights and predictive understanding is weak. With such detailed information, it should be possible to provide some guideline to build quantifiable prediction model for particle degradation.

2. The logical flow in the following paragraphs is vague: the one starting at line #119 and the one starting at line #200.

3. In the introduction, there are few misconstrued statements in regards to the literature which are factually wrong. For example, in line 47, the referred paper says exactly the opposite as claimed by the authors.

Reviewer #2 (Remarks to the Author):

NCOMMS-19-33161-T is an interesting manuscript that reports in-situ x-ray diffraction (XRD) tomographic study of a $\text{Li}_x\text{Mn}_2\text{O}_4$ electrode as it was electrochemically cycled. The cycling procedure was purposely paused for the data acquisition to take place because the measurements are rather time-consuming. Although this material system is already well investigated and documented, the authors nicely utilized the XRD tomography's sensitivity to the local crystallographic structure to visualize the inter and intra particle chemical heterogeneity, which is a broadly observed phenomenon and is believed to play an important role in the cathode degradation. In particular, the authors captured many particles at once with micrometer spatial resolution, which facilitated in-depth analysis of two selected particles, highlighting the coexistence of multiple phases within individual particles. I think this manuscript is of interest to researchers working in this field and, therefore, I recommend the publication of this paper with specific comments listed below for the authors to address.

1) The authors highlighted that their data has five dimensions (x, y, z, time, and diffraction). However, the majority of the manuscript focused on a single slice, which is understandable as the multi-slice data acquisition would take much too long for a study like this. I would recommend the authors to take out the buzzword "5D", which doesn't add much scientific value here, instead, it could cause confusion.

2) The electrode used in this work was 80 microns thick. One would expect to see a significant polarization effect in such a configuration. [see Adv. Energy Mater. 1900674, (2019)]. How did the authors choose the z position? Some discussions along this line will be useful.

3) Maybe I missed this point, how was the reconstruction carried out? Did the authors conduct the projection XRD data refining, then use the phase fractions as input for tomographic reconstruction?

4) The authors nicely captured many particles at once, which could offer an opportunity for statistical analysis. For example, is there any particle size dependence? A quick look at Figure 3a suggests that smaller particles appear to be red-orange, and the larger particles appear to be pink. I understand that the actual available data is over a 2D slice, but couldn't help to notice this pattern.

5) Particle 1 is attributed to the Li-rich compound $\text{Li}_{1.10}\text{Mn}_{1.90}\text{O}_4$. Where is it coming from? It has been reported that the resistance for Li diffusion in the Li-rich compound could be quite high at charged state [RSC Adv. 2, 8797 (2012); J. Phys. Chem. C 114, 22751, (2010)]. It could result in a core-to-surface Li concentration gradient [J. Am. Chem. Soc. 141, 30, 12079-12086 (2019)]. How do

these reports relate to the observation in this current work?

Reviewer #3 (Remarks to the Author):

This is a strong submission. Questions for the authors:

On line 78 the authors detect crack formation starting from internal pores. Does this support or contradict the predictions of Cheng et al, Journal of Materials Research 25:1433 (2010)?

On line 88, "inter" should probably be "intra." Do the authors have any information about inter-particle cracks, such as would occur when the binder fails?

I don't understand the significance of the arrow in Figure 2a. Is it meant to show that a particular particle observed in the first image was imaged again in image 1? If so, where is the particle in the other images?

I'm not sure about the description of some of the histograms in Figure 2a as bimodal. It could be that the authors are referring to the dip in the center of the histogram. If so, I am not convinced that the dip is real and not just noise.

On line 177 the authors state that the cell was cycled twice. Do they have data for each cycle? Are they significantly different?

In previous work the authors have seen macroscale strains during lithiation. Do they see it here also?

Reviewer #1 (Remarks to the Author):

The article is based on solid experimental results consisting of high resolution XRD-CT data. The data provide the framework to understand the phase change and degradation mechanisms in LMO cathode particles and can be extended to other material systems. In this respect, the manuscript is valuable. However, I have some reservations regarding its publication in Nat-Comm as mentioned below.

We sincerely thank the reviewer for their time in reviewing our work and for providing their comments. We are glad that the reviewer considers our work solid and valuable, and hope that we assuage their reservation in our response below.

1. The logical argumentation which provides the extension of experimental observations to the insights and predictive understanding is weak. With such detailed information, it should be possible to provide some guideline to build quantifiable prediction model for particle degradation.

We appreciate the reviewers comment on improving the logic of our argument and building a quantifiable prediction model. From the data that we have, we are able to draw some insight into the degradation mechanism of LMO, but as the reviewer highlights, we cannot complete a full quantifiable prediction model for particle degradation. At minimum, this would require periodic high-resolution XRD-CT with many slices of the same cell during numerous cycles, which is not feasible with current synchrotron and data processing capabilities.

However, we have made a significant contribution to the grand objective of building a quantifiable prediction model and we have also demonstrated the efficacy of XRDCT in probing sub-particle heterogeneities and drawing quantitative phase fraction comparisons within and between particles, which is the first demonstration of its kind.

For example, we know from previous literature that Mn dissolution is connected to accelerated degradation of capacity and performance, and we know that phases Li_2MnO_3 and LiMnO_2 have negative effects on the cell's performance, but before this work we did not have any spatial understanding of how and where they begin to form, which is a crucial step towards building a quantifiable prediction model. Here, we show that degradation products are already present in a small fraction of the particles from the pristine state (e.g. particle 1 showing Li_2MnO_3) thus residual defects from the synthesis step cannot be neglected when considering electrode degradation. We also show that large amounts of LiMnO_2 forms locally within particles upon the first discharge thus affecting the diffusion coefficient and interfacial reaction barrier early in the cell's life, thus assuming the diffusion coefficient and interfacial properties of pure LMO in a model may not be accurate. From the degraded sample presented in our work, we also show extreme heterogeneities in degradation with some regions showing up to 20% composition of LiMnO_2 and 11% Li_2MnO_3 and other regions showing compositions that are almost akin to the fresh electrode with only 6% Li_2MnO_3 , and even though we cannot definitively say how such localised compositions came to be, this insight does show that the rate of degradation, and

perhaps degradation mechanism, is a local phenomena which is another important consideration for a complete degradation model and a worthy topic for future work.

However, as per the reviewer's comment, we acknowledge that perhaps the structuring of our manuscript could be improved to convey these points in a more logical way. Therefore, we have made the following modifications:

- We have restructured the abstract to emphasize the insights of inter and intra particle heterogeneous behaviour.

- We have further emphasized the heterogeneity angle in the introduction:

E.g. "It is shown how the stoichiometry of LMO and its crystallographic response to lithiation varies between particles during operation, and how the stoichiometry of distinct particles changes due to Mn dissolution upon extensive cycling, the extent of which varies widely between particles."

- We have provided more literature and discussion on the impact of LiMnO_2 phases in the second paragraph following figure 4:

E.g. "The behavior of Particle 2 is akin to the bulk electrode, hence the formation of LiMnO_2 in such quantities as observed in Figure 5d of up to 26%, could have significant consequences for the performance of the electrode. For example, as examined by Yu et al.⁵², LiMnO_2 has a relatively low diffusion coefficient and high interfacial reaction barrier thus indicating that the presence of this phase in the bulk electrode shown here may negatively affect its rate performance."

- Finally, we restructured the conclusion to improve clarity on the importance of this work in understanding the degradation of LMO as well as the opportunities that are presented from demonstrating the power of this technique.

2. The logical flow in the following paragraphs is vague: the one starting at line #119 and the one starting at line #200.

We thank the reviewer for bringing this to our attention. Upon revision, we agree that the two paragraphs could be phrased differently to improve clarity and coherence. We have modified the paragraphs as follows:

Paragraph at line 119:

"The XRD-CT slices in Figure 2a show that there are inter and intra particle lattice parameter heterogeneities, the distributions of which are presented as histograms in Figure 2b for each of the XRD-CT slices. With knowledge from previously published literature, the shape and evolution of the lattice parameter histograms can provide insight into the phase heterogeneities that existed and evolved upon lithiation. When lithiating, $\text{Li}_x\text{Mn}_2\text{O}_4$ is known to undergo two bi-

phasic reactions^{24, 25}, the first of which consists of a transition from a phase with lattice parameter $a \approx 8.076 \text{ \AA}$ directly to a phase with $a \approx 8.145 \text{ \AA}$ ^{24, 45}. The second phase change is relatively slight, transitioning from $a \approx 8.183 \text{ \AA}$ directly to $a \approx 8.205 \text{ \AA}$. The lattice parameter range over which the first (larger) phase change occurs is highlighted as pink in Figure 2b, and is a range that we would not expect to be occupied if the system were pure spinel $\text{Li}_x\text{Mn}_2\text{O}_4$. However, some volume of the electrode was observed to have occupied this range, thus exhibiting lattice parameter values that did not correlate with the characteristic behavior of the spinel $\text{Li}_x\text{Mn}_2\text{O}_4$ stoichiometry. From the sequence of XRD-CT slices in Figure 2a, the particles that were observed to consistently deviate from the bulk behavior of $\text{Li}_x\text{Mn}_2\text{O}_4$ were the ones that occupied the non-characteristic range highlighted in pink in Figure 2b. To understand the discrepancy between the non-characteristic particles and the bulk electrode, single particle analyses were carried out for further insight.”

Paragraph at line 200:

“The formation of LiMnO_2 and Li_2MnO_3 phases during cycling is well-documented^{29, 49, 50}. Their presence can stem from dissolution of Mn^{2+} from spinel $\text{Li}_x\text{Mn}_2\text{O}_4$ and the consequent oxidation of the residual Mn^{3+} or Mn^{4+} phases. The resulting LiMnO_2 and Li_2MnO_3 are more thermodynamically stable and co-exist with the $\text{Li}_x\text{Mn}_2\text{O}_4$ phase. Migration and deposition of the dissolved Mn on the graphite electrode can cause severe capacity fade through consumption of Li and impedance-rise^{9, 51}. Here, evidence for Mn dissolution was found by examining the materials at both the positive and negative electrodes. At the graphite negative electrode, X-ray fluorescence confirmed Mn deposits following suspected dissolution and migration of Mn through the electrolyte (see Section 2 of Supplementary Information)⁹. At the positive electrode, regions with lowest lattice parameter (Segmentation 1 in Figure 6) were shown to have contained high mass fractions of products that arise from Mn dissolution with 11 % Li_2MnO_3 and 20 % LiMnO_2 . The lattice parameter for the LiMnO_2 phase was 4.13 \AA which is similar to that found by Tu and Shu²⁹. From Figure 6, it is seen that the higher the fraction of segregated phases (Li_2MnO_3 and LiMnO_2), the lower the lattice parameter values observed; this is explained by the fact that as the Mn in $\text{Li}_x\text{Mn}_2\text{O}_4$ undergoes disproportionation, all product spinel compounds will have lower lattice parameter values than the original $\text{Li}_x\text{Mn}_2\text{O}_4$ phase due to the original spinel having the highest concentration of the relatively large Mn^{3+} ion⁴⁹.

With co-existence of phases within single particles, internal strains are expected to have formed. A strain analysis on the three segmented regions in Figure 6c,d was carried out and is shown in Section 9 of Supplementary Information. The strain analysis showed that like Particle 2, strain was highest for regions that had the greatest fraction of segregated phases i.e., highest strain for regions that experienced the highest degree of Mn dissolution. The gained insight into

the evolution of sub-particle phase segregation during cycling, and the increase in fraction of segregated phases over many cycles, may help explain the onset of the cracks observed in SEM images in Figure 1c,d which were taken from the same cycled sample.”

3. In the introduction, there are few misconstrued statements in regards to the literature which are factually wrong. For example, in line 47, the referred paper says exactly the opposite as claimed by the authors.

We thank the reviewer for bringing this to our attention. We are aware that the mechanisms through which Mn is lost from LMO are not yet fully understood and that our statement

“the consensus for the mechanisms of Mn dissolution is that a disproportionation reaction takes place”

was poorly phrased and not accurate for the following reason: There is a consensus that disproportionation is one mechanism in which Mn dissolution can occur BUT not the only one, as discussed in the referred paper by Bhandari and Battacharya and summarized in their concluding statement

“The disproportionation hypothesis certainly does not explain all the observations in the literature. In the above light, it appears that disproportionation hypothesis may be a sufficient condition for dissolution but need not be a necessary one”

We have rephrased our sentence to better reflect this uncertainty, and have provided reference to Bhandari’s paper for readers to follow for more information on the uncertainties surrounding a complete description of Mn dissolution processes:

“One mechanism of Mn dissolution but perhaps not the only one²³, is that a disproportionation reaction takes place at the interface of the LMO and electrolyte where Mn sites separate into Mn^{2+} and Mn^{4+} , where the Mn^{2+} dissolves²³. “

Finally, the reviewer used the plural “a few misconstrued statements”. We have read over the introduction, and could not identify any others.

Reviewer #2 (Remarks to the Author):

NCOMMS-19-33161-T is an interesting manuscript that reports in-situ x-ray diffraction (XRD) tomographic study of a $LixMn_2O_4$ electrode as it was electrochemically cycled. The cycling

procedure was purposely paused for the data acquisition to take place because the measurements are rather time-consuming. Although this material system is already well investigated and documented, the authors nicely utilized the XRD tomography's sensitivity to the local crystallographic structure to visualize the inter and intra particle chemical heterogeneity, which is a broadly observed phenomenon and is believed to play an important role in the cathode degradation. In particular, the authors captured many particles at once with micrometer spatial resolution, which facilitated in-depth analysis of two selected particles, highlighting the coexistence of multiple phases within individual particles. I think this manuscript is of interest to researchers working in this field and, therefore, I recommend the publication of this paper with specific comments listed below for the authors to address.

We express our thanks to the reviewer for reading our work and for conducting a thorough review. We appreciate their constructive comments and recommendation, and have provided our response to their comments below.

1) The authors highlighted that their data has five dimensions (x, y, z, time, and diffraction). However, the majority of the manuscript focused on a single slice, which is understandable as the multi-slice data acquisition would take much too long for a study like this. I would recommend the authors to take out the buzzword "5D", which doesn't add much scientific value here, instead, it could cause confusion.

We agree with the reviewer and have removed "5D" to avoid confusion.

2) The electrode used in this work was 80 microns thick. One would expect to see a significant polarization effect in such a configuration. [see Adv. Energy Mater. 1900674, (2019)]. How did the authors choose the z position? Some discussions along this line will be useful.

We thank the reviewer for bringing this recent study to our attention which highlights that after 10's of cycles at high rate (5C), degradation of cathode particles can be depth-dependent with most severe degradation being closest to the separator. Depth-dependent heterogeneities are actually a topic of a separate study that we will soon complete on anode materials. However, in this work we operated at C/4 for the degraded sample that was cycled 150 times, and C/1.8 for the operando sample that was only cycled once with intermittent stops for imaging. In the degraded sample, we did not consider depth-dependency of degradation to be significant at C/4 rate, hence we just took one slice from the middle of the electrode and reasoned that at such low rates the degradation would be somewhat homogeneous with depth. However, we cannot be certain of this and so cannot say for sure that our observations are uniform for all depths, thus this is an important point to discuss in our manuscript.

We have clarified the location at which the XRDCT slices were taken, have included the reference provided by the reviewer [Adv. Energy Mater. 1900674, (2019)], and have included a short discussion on this point:

First paragraph in section 2.2:

“An initial XRD tomogram with a volume of $202\ \mu\text{m} \times 202\ \mu\text{m} \times 40\ \mu\text{m}$, with $2\ \mu\text{m}$ vertical and horizontal spatial resolutions, was acquired midway through the electrode’s depth to identify a region of interest for further high resolution scans (Figure 2a). Thereafter, $301\ \mu\text{m} \times 301\ \mu\text{m} \times 1\ \mu\text{m}$ XRD-CT slices were acquired close to mid-way through the electrode depth at different stages during discharge of the cell (lithiation of LMO) in which distinct particles could be identified (Figure 2a).”

First paragraph of section 2.4:

“An XRD-CT slice was taken from around mid-way through the depth of the electrode. It has previously been shown that degradation can vary with depth into the electrode^{53, 54}, but here we did not focus on depth-dependency due us not expecting significant gradients in lithiation conditions at the low operating rate of $C/4$. However, we cannot say for certain that the degradation conditions observed for a slice mid-way through the electrode were representative of all depths. “

53. Yang, Y. et al. Quantification of Heterogeneous Degradation in Li-Ion Batteries. **9**, 1900674 (2019).

54. Yao, K.P.C., Okasinski, J.S., Kalaga, K., Shkrob, I.A. & Abraham, D.P. Quantifying lithium concentration gradients in the graphite electrode of Li-ion cells using operando energy dispersive X-ray diffraction. *Energy & Environmental Science* **12**, 656-665 (2019).

3) Maybe I missed this point, how was the reconstruction carried out? Did the authors conduct the projection XRD data refining, then use the phase fractions as input for tomographic reconstruction?

No, as the approach that the reviewer mentions is not generally valid for inhomogeneous samples. The reviewer is correct that one option is to perform peak fitting using the XRD projection data and then reconstruct features that contain physical or chemical information (e.g. phase distribution maps). However, one has to be careful about this approach as there can be areas in the sample where a specific phase is nanocrystalline, therefore generating very broad diffraction peaks, while in other areas the same phase may be highly crystalline, leading to the formation of very sharp diffraction peaks. In such a case, the peaks should probably be treated as a two-phase problem, otherwise it is impossible to apply a correct peak-shape function to fit the data (what is an appropriate FWHM to use?). However, if there is a distribution of crystallite sizes, then the peak-fitting process becomes more challenging. An alternative option to obtain the reconstructed images is the *reverse analysis* method where the whole projection data set volume is reconstructed, leading to a $T \times T \times d$ matrix (i.e. a three-dimensional matrix). In this work, we chose the *reverse analysis* method where every pixel in the reconstructed XRD-CT image contains or corresponds to a single diffraction pattern (Bleuet *et al.*, 2008). We then performed the analysis (Rietveld, Williamson-Hall etc) using the reconstructed diffraction patterns. These patterns are easier to interpret as they contain real-space local physico-chemical information and require the use of a single model (in contrast to the projection data).

4) The authors nicely captured many particles at once, which could offer an opportunity for statistical analysis. For example, is there any particle size dependence? A quick look at Figure 3a suggests that smaller particles appear to be red-orange, and the larger particles appear to be pink. I understand that the actual available data is over a 2D slice, but couldn't help to notice this pattern.

The reviewer raises an interesting suggestion of carrying out a statistical analysis of, for example, examining any relationships between particle size and composition. The reviewer is correct in having caution due to the data being from one single slice, where a particle with a small diameter may actually just be the tip of a larger particle which would be observed with full 3D information. We did not capture full 3D information due to time-constraints (each slice took about 45 mins) so with the existing data, we cannot confidently say that a small diameter observed in our 2D data is actually a small particle volumetrically, therefore with the existing data a statistical analysis on particle size is not possible. However, The European Synchrotron (ESRF) is undergoing an upgrade to be the Extremely Brilliant Source (EBS) which is expected to significantly increase the speed of acquisition for similar future studies, e.g. it might be possible to achieve a large 3D sample size (e.g. 150 x 150 x 30 μm) with 1 μm resolution in about 30 mins. We have had discussions with the ESRF beamline scientists about this. Hence, we consider this important to discuss in the Conclusions section of the manuscript. We have added the following to the Conclusion:

“However, the time required for acquisition of the data presented in this work limited analyses to a single slice and thus prevented statistical volumetric comparisons such as degradation as a function of particle size. The ongoing Extremely Brilliant Source (EBS) upgrade at the ESRF is expected to dramatically decrease acquisition time, and perhaps facilitate operando volumetric comparisons for large samples sizes in future work. “

5) Particle 1 is attributed to the Li-rich compound $\text{Li}_{1.10}\text{Mn}_{1.90}\text{O}_4$. Where is it coming from? It has been reported that the resistance for Li diffusion in the Li-rich compound could be quite high at charged state [RSC Adv. 2, 8797 (2012); J. Phys. Chem. C 114, 22751, (2010)]. It could result in a core-to-surface Li concentration gradient [J. Am. Chem. Soc. 141, 30, 12079-12086 (2019)]. How do these reports relate to the observation in this current work?

We thank the reviewer for bringing this work to our attention. We think that the Li-rich compound is formed due to heterogeneous synthesis conditions, i.e. in the batch synthesis process, some particles take on a slightly different stoichiometry than others. The paper by Yu et al. (RSC Adv. 2, 8797 (2012)) is valuable for furthering insight into the function of our LMO electrode since we observe significant amounts of LiMnO_2 forming in the bulk electrode (e.g. see figure 5). The paper by Yu et al. gives insight into the relatively poor performance of LiMnO_2 and thus has consequences for the bulk electrode's performance in our work. Hence, we have added the following and cited the paper in the paragraph following Figure 4:

“The behavior of Particle 2 is akin to the bulk electrode, hence the formation of LiMnO_2 in such quantities as observed in Figure 5d of up to 26%, can have significant consequences for the

performance of the electrode. For example, as examined by Yu et al.⁵⁰, LiMnO₂ has a relatively low diffusion coefficient and high interfacial reaction barrier thus indicating that the presence of this phase in the bulk electrode shown here may negatively affect its rate performance.”

As for the core-to-surface Li concentration gradient in (J. Am. Chem. Soc. 141, 30, 12079-12086 (2019)) - it is possible that the observations in this paper are related to what we see for the anomalous particle in our manuscript. It is difficult to make a direct comparison with this work due to the operating environments being very different, but the referred paper does provide a good example of how single particles can have Li concentration gradients and thus we have added the reference to support our work:

“The gradient in Particle 1 consisted of the highest lattice parameter in the center of the particle, for which the most likely explanation is that the particle itself had a slight stoichiometric or phase gradient⁴⁹”

49. Li, S. et al. Surface-to-Bulk Redox Coupling through Thermally Driven Li Redistribution in Li- and Mn-Rich Layered Cathode Materials. *Journal of the American Chemical Society* **141**, 12079-12086 (2019).

Reviewer #3 (Remarks to the Author):

This is a strong submission. Questions for the authors:

We thank the reviewer for their time spent reviewing our work and for their positive remark on the strength of our manuscript. We have provided answers for the reviewer's questions below.

On line 78 the authors detect crack formation starting from internal pores. Does this support or contradict the predictions of Cheng et al, *Journal of Materials Research* 25:1433 (2010)?

We thank the reviewer for bringing this paper to our attention. The nature of the cracks in Figure 1 appears to support the predictions in the referred paper. In section 2.1, we have added this reference to provide a source for further insight into the observations in Figure 1:

“The same electrode was cycled 150 times (see Experimental Section for details) and as predicted by Woodford et al.[41], larger particles exhibited a greater tendency to crack, and cracks tended to stem from the edges of internal pores (Figure 1c-d) as previously predicted[42].”

41. Woodford, W.H., Chiang, Y.-M. & Carter, W.C. “Electrochemical Shock” of Intercalation Electrodes: A Fracture Mechanics Analysis. *Journal of The Electrochemical Society* **157**, A1052-A1059 (2010).

42. Harris, S.J., Deshpande, R.D., Qi, Y., Dutta, I. & Cheng, Y.-T. Mesopores inside electrode particles can change the Li-ion transport mechanism and diffusion-induced stress. *Journal of Materials Research* **25**, 1433-1440 (2010).

On line 88, "inter" should probably be "intra." Do the authors have any information about inter-particle cracks, such as would occur when the binder fails?

We thank the reviewer for identifying this typo. We have corrected "inter" to be "intra".

For the reviewers information, we have explored inter particle cracks in a previous publication [see Finegan et al. <https://doi.org/10.1002/adv.201500332>] where we investigated the relationship between electrode motion during lithiation and the formation of intercracks within the cathode of a primary Li vs LMO commercial cell. However in this work, we did not explore inter-particle cracks as it is beyond the scope of the study.

I don't understand the significance of the arrow in Figure 2a. Is it meant to show that a particular particle observed in the first image was imaged again in image 1? If so, where is the particle in the other images?

The purpose of the first image was to quickly capture a larger, but coarser, 3D view of the system to determine whether there were any anomalies worth homing in on. In other words, it was a low-resolution scan to first determine a region of interest (ROI). From the technique perspective, this method of fast determination of an ROI from a coarse scan is quite powerful, and to the authors' knowledge, this is the first demonstration of its kind for XRDCT of a Li-ion electrode.

In the other images (1-5), the particle is in the same location but in images 2 and 3 it is difficult to distinguish because it takes on a similar lattice parameter value (and color) to its neighbors. In images 4 and 5, the particle stands out again (orange-ish).

I'm not sure about the description of some of the histograms in Figure 2a as bimodal. It could be that the authors are referring to the dip in the center of the histogram. If so, I am not convinced that the dip is real and not just noise.

After further consideration, we agree with the reviewer and have removed the following sentence that refers to the bimodal feature "A bimodal distribution of lattice parameter forms in XRD-CT 3 to XRD-CT 5 indicating that a substantial fraction of the electrode underwent phase segregation."

On line 177 the authors state that the cell was cycled twice. Do they have data for each cycle? Are they significantly different?

The "cycled twice" was a typo pasted from the pouch cell description. The operando cell was charged once before the experiment, hence its discharge was from a pristine lithiated state. The

typo has been corrected in the manuscript and the charge data has been added to the supplementary material and is shown below.

The charge profile is characteristic of an LMO vs. Li cell and shows that the operando cell design was functioning as desired when outside the beam. The discharge profile was included in the manuscript and a version with the OCV periods subtracted was also included in the Supplementary Information. Both discharge profiles (with and without OCV) are provided below for comparison. As seen, when the cell is put into the beam, the characteristic plateaus of LMO were not as obvious, but this may also be due to the higher rate of C/1.8.

Charge profile outside of beam:

Discharge profile inside the beam (already included in the Supplementary Information):

Figure S5: (a) Voltage [black] and current [red] profile of the micro-cell during the XRD-CT experiment. The XRD-CT images were taken during the open circuit periods. (b) The voltage profile with respect to $\text{Li}_x\text{Mn}_2\text{O}_4$ for the micro-cell with the open circuit periods removed [black] and a standard coin cell [gray]. The micro-cell contained Li as the counter electrode, whereas the coin cell contained graphite, which helps explain the discrepancy in voltage between the two.

In previous work the authors have seen macroscale strains during lithiation. Do they see it here also?

We agree with the reviewer and indeed recognise that strain analysis is very important in battery/electrode systems. If the reviewer is referring to mechanical (displacement) strain of the electrode-coating (as measured by Finegan et al using DVC in <https://doi.org/10.1002/adv.201500332>), we have not measured this, as this required full 3D information and is beyond the scope of this study. However, if the reviewer is referring to crystallographic micro-strain analysis within the bulk electrode and distinct particles, the answer is yes but with some caveats, as described below.

We performed a micro-strain analysis using the diffraction data from the bulk electrode but it might have been difficult to be noticed as it was at the very end of the Supporting Information (Section 9. Strain analysis for the LiMn_2O_4 phase). In this work, the LiMn_2O_4 diffraction peaks

(the main crystalline phase present) were sharp (as indicated by the various diffraction patterns presented throughout the manuscript) and the peak shape did not alter significantly during the in situ experiment. We used the Williamson-Hall approach to decouple the peak broadening contributions associated with crystallite size/ strain. The “good” particle (Particle 2) was selected as it was representative of the bulk electrode, and the summed diffraction pattern from this particle for the five XRD-CT datasets was used to explore the strain.

It was seen that the changes in the Williamson-Hall plot between the respective XRDCT scans, were predominantly related to the slope rather than the intercept of the fitted lines. This suggests that there was an increase in strain for the LiMn_2O_4 phase as it lithiated, which could have been caused by a variety of reasons including phase segregation (the formation of the LiMnO_2 rock-salt phase as observed in XRD-CT dataset 2) and the lithiation process itself. It should be noted that there was an increase from XRD-CT dataset 1 to XRD-CT dataset 2 which was coincident with the LiMnO_2 rock-salt phase formation. The fits were worse for XRD-CT datasets 4 and 5 as the points deviate more from the linear model implying the presence of other features in the LiMn_2O_4 peaks (this coincident with Li occupancy exceeding 1). We also performed the same analysis for the cycled cell and the results are in agreement with results from the in situ experiment as the LiMn_2O_4 strain is higher in the regions where the various undesired phases, such as the Li_2MnO_3 and the LiMnO_2 cubic rock-salt, are present.

During this revision, we attempted to perform this analysis in a spatially-resolved manner, rather than just taking bulk measurements as described in the previous paragraph. As the signal-to-noise ratio was worse for the spatially-resolved diffraction patterns present in the reconstructed XRD-CT data, only the first seven LiMn_2O_4 diffraction peaks were used for the analysis because they were the most distinct. We developed in-house python code for batch multi-peak fitting using the `scipy.optimize` package and used gaussian peak shapes (<http://pd.chem.ucl.ac.uk/pdnn/peaks/gauss.htm>). Initially, the peak fitting using the same code was performed using the CeO_2 pattern and the full-width-at-half-maximum (FWHM) of the peaks was calculated. The 2θ broadening of the peaks was modeled using the Caglioti formula and refining the U, V and W parameters. The batch peak fitting was performed using the Gaussian peak shape and the instrumental broadening (calculated using the Caglioti formula for the LiMn_2O_4 peak positions) was subtracted from the obtained FWHM values (<http://pd.chem.ucl.ac.uk/pdnn/peaks/broad.htm>) before performing the first degree polynomial fitting for the Williamson-Hall plots (FWHM*cos(theta) vs sin(theta)). The results obtained from this spatially-resolved analysis are presented in the following Figure. It can be seen that the results are in agreement with the ones obtained from the representative particle (Figure S36 in Supplementary Information) as it is shown that the slope (corresponding to strain) increased from XRD-CT dataset 1 to 5 with a small decrease in intercept (corresponding to crystallite size). However, one should be careful when interpreting these results as although the Rwp values from the peak fitting are low, the error in the slopes and intercepts is more significant (see norm of the residuals in Figure below (this has been added to Supplementary Information)) and the number of peaks used can have a strong impact on the obtained values. For these reasons, we maintained our analysis of the cycled cell using the summed diffraction patterns from the three segmented regions (Figure 37) rather than performing the spatially-resolved analysis.

So in conclusion, we have observed an increase in strain with lithiation across the bulk electrode but we cannot definitively state the exact cause of this strain although it appears to correlate with phase segregation. We also observed spatially resolved strain, but the error in such localised values was high. We have added the new figure to the Supporting Information.

REVIEWERS' COMMENTS:

Reviewer #1 (Remarks to the Author):

The queries have been addressed adequately. I recommend acceptance in the present form.

Reviewer #2 (Remarks to the Author):

The authors have addressed my comments nicely.
I recommend the publication of this manuscript in its present form.

Reviewer #3 (Remarks to the Author):

Publish as revised